# Coronary Flow and Reserve by Enhanced Transthoracic Doppler Trumps Coronary Anatomy by Computed Tomography in Assessing Coronary Artery Stenosis

**DOI:** 10.3390/diagnostics11020245

**Published:** 2021-02-05

**Authors:** Carlo Caiati, Arnaldo Scardapane, Fortunato Iacovelli, Paolo Pollice, Teresa Immacolata Achille, Stefano Favale, Mario Erminio Lepera

**Affiliations:** 1Institute of Cardiovascular Disease, Department of Emergency and Organ Transplantations, University of Bari “Aldo Moro”, 70124 Bari, Italy; fortunato.iacovelli@gmail.com (F.I.); paolo.pollice@yahoo.it (P.P.); achille_teresa.i@libero.it (T.I.A.); stefano.favale@uniba.it (S.F.); marioerminio.lepera@uniba.it (M.E.L.); 2Interdisciplinary Department of Medicine, Section of Diagnostic Imaging, Bari Medical School, 70124 Bari, Italy; arnaldo.scardapane@gmail.com

**Keywords:** coronary stenosis, coronary computed tomography, coronary transthoracic doppler echocardiography

## Abstract

We report the case of a 71-year-old patient with many risk factors for coronary atherosclerosis, who underwent computed coronary angiography (CTA), in accordance with the guidelines, for recent onset atypical chest pain. CTA revealed critical (>50% lumen diameter narrowing) stenosis of the proximal anterior descending coronary, and the patient was scheduled for invasive coronary angiography (ICA). Before ICA he underwent enhanced transthoracic echo-Doppler (E-Doppler TTE) for coronary flow detection by color-guided pulsed-wave Doppler recording of the left main (LMCA) and whole left anterior descending coronary artery (LAD,) along with coronary flow reserve (CFR) in the distal LAD calculated as the ratio, of peak flow velocity during i.v. adenosine (140 mcg/Kg/m) to resting flow velocity. E-Doppler TTE mapping revealed only mild stenosis (28% area narrowing) of the mid LAD and a CFR of 3.20, in perfect agreement with the color mapping showing no flow limiting stenosis in the LMCA and LAD. ICA revealed only a very mild stenosis in the mid LAD and mild atherosclerosis in the other coronaries (intimal irregularities). Thus, coronary stenosis was better predicted by E-Doppler TTE than by CTA. Coronary flow and reserve as assessed by E-Doppler TTE trumps coronary anatomy as assessed by CTA, without exposing the patient to harmful radiation and iodinated contrast medium.

## 1. Introduction

Coronary artery disease is an escalating disease and the leading cause of death in westernized countries so early, non-invasive detection is utterly vital in the management of the disease [1].

One emerging diagnostic approach is computed tomography coronary angiography (CTA). This approach is non-invasive but exposes the patient to the risk of ionizing radiation [2] and also to iodinated contrast medium [3]. Moreover, CTA allows only a luminal evaluation of the severity of a stenosis, that is already a limitation, but it is compounded by the lower resolution of the method compared to invasive coronary angiography, the gold standard for luminology. Hence, the risk is an even more imprecise evaluation of coronary stenosis that could severely hamper clinical decision making as false positives can emerge [4].

Trans-thoracic enhanced Doppler echocardiography (E-Doppler TTE) can assess the blood flow velocity in the left main coronary artery (LMCA) and the whole left anterior descending coronary artery (LAD). It has recently been demonstrated to have very high feasibility, thanks to the combination with three technological advances: new power Doppler-based technology (convergent color-Doppler mode), along with pharmacologically lowering the heart rate, and new tomographic planes [5]. This method allows the assessment in these two vessels primarily of the transtenotic coronary flow velocity that is usually expressed as percentage velocity increment with respect to a reference site. Such a velocity increment is an adaptive mechanism to compensate for lumen narrowing in order to keep the flow constant [6]. It can be accurately measured by E-Doppler TTE thanks to the very recent technical advances [5], and the severity of the stenosis can be assessed by applying the continuity equation [5]. Moreover, it allows the assessment of the distal coronary flow reserve (CFR), a major parameter for assessing global coronary physiology [7,8,9,10]. The acceleration of blood flow is crucially important in demonstrating a low CFR due to critical stenosis and not to a microcirculation dysfunction, as recently preliminarily demonstrated for the first time [11]. Moreover, this result is obtained without exposing the patient either to ionizing radiation or iodinated contrast medium.

For a first time, in a real clinical scenario, we report the comparison of E-Doppler TTE with CTA in a subject with recent onset atypical chest pain. CTA showed obstructive atherosclerosis in the proximal-mid LAD that was quantified as critical, prompting the performance of invasive coronary angiography (ICA). E-Doppler TTE, performed before ICA based on functional parameters, showed a very mild stenosis in the mid LAD with a CFR in the distal LAD of 3.2, that confirmed the non-limiting stenosis flow in the LMCA and LAD. The findings by E-Doppler TTE were confirmed by ICA. Thus, coronary flow in the LMCA and LAD assessed by E-Doppler TTE trumps coronary anatomy assessed by CTA, without exposing the patient to harmful ionizing radiation.

## 2. The Case

A 71-year-old man with important risk factors for coronary artery disease (CAD)—systemic hypertension, a previous heavy tobacco smoking habit and a strong familial predisposition to CAD (his father died of acute myocardial infarction at the age of 50 years) developed atypical chest pain. Two years before he had experienced an occlusion of the right internal carotid artery that provoked monolateral amaurosis. On the basis of his symptoms and in accordance with the European Society of Cardiology (ESC) guidelines [12], his cardiologist suggested a CTA scan, that the patient underwent as an outpatient. Before CT acquisition the patient received 0.15 mg of Trinitrine in a sublingual spray. Because the cardiac frequency was higher than 70 bpm, a dose of 5 mg of Metoprolol was administered intravenously immediately before imaging but the heart rate reduction obtained was suboptimal, at >65 b/m. CTA was acquired using a 160 slice multirow detector scanner (Aquilion Premium, Toshiba Medical System, Tustin, CA, USA); modulated ECG gated images were obtained with the following protocol: slice thickness 0.6 mm, image index 0.4 mm, pitch factor “detailed”, mAs 250, kV 100 scan time 5–7 s). A high concentration of iodinated contrast medium (400 mg/I/mL) was administered through a cubital vein with a 16 G cannula and an automatic power injector (volume 50 mL, flow rate 5 mL/s). The scan delay was assessed with a bolus tracking series, placing a region of interest (ROI) in the descending thoracic aorta and setting a cut-off density of 110 HU. All the images were transferred onto a dedicated workstation (VITREA Fx, Vital Images), where automated segmentation of the heart was performed and curved multiplanar reformation (MPR) images of coronary arteries were reviewed. This CTA reading showed a critical multivessel obstructive CAD (Figure 1), with stenosis provoking >50% of lumen diameter narrowing, involving both the mid LAD (Figure 1), and a first diagonal and the mid right coronary artery (RCA). In addition, a possible atherosclerotic involvement of the circumflex coronary artery (LCx) was described.

### Hospital Admission

On the basis of this result and in particular the involvement of the LAD, he was admitted to our tertiary hospital for invasive coronary angiography and revascularization if necessary. He was taking the following drugs: Olmesartan 20 mg, Allopurinol 150 mg; Clopidogrel 75 mg and Atorvastatin 40 mg. He was mildly overweight. Physical tests were unremarkable: the apex beat was normal in terms of location (<10 cm from the mid sternal line) and duration (the outward trust was felt only in the first third part of the systole and faded away well before the second heart sound); no palpable presystolic extra hump was found. The venous systemic pressure was normal and with a good X’ descent (indicating good right ventricular function and normal ventricular-pulmonary artery coupling); carotid palpation indicated normal rate of rise and normal amplitude. The auscultation was normal, confirming the pulses and apex beat data. The EKG and chest X-ray were unremarkable and creatinine value was 1.07 mg/dL. 

In the hospital, before ICA he underwent E-Doppler TTE. This exam was performed using an Acuson Sequoia™ ultrasound unit (C256 Echocardiography System, Siemens Healthcare, Erlangen, Germany) and broadband transducer (3V2c). The color Doppler signal was attained in convergent color-Doppler mode at 2.5 MHz transmission frequency, while spectral Doppler was performed in fundamental mode at either 2.0 or 2.5 MHz. The color-coded Doppler setting was adjusted to maximize scanning sensitivity (pulse repetition frequency was set at 16 cm/s (2.5 MHz) and maximizing the sample volume of color flow mapping) without significantly reducing the frame rate (the color box size was reduced to remain within a frame rate of >30 Hz) [5].

Before the examination, the heart rate was reduced with beta blockers (100 mg of metoprolol PO) to a suboptimal value of 65 b/m [5]. He had refrained from drinking beverages containing caffeine already three days before the hospital admission. He was overweight so his standard echocardiography window was limited. Owing to obesity and the suboptimal heart rate reductions, the examination was of suboptimal quality but nonetheless fully diagnostic (this kind of examination has a feasibility of ≅100%, as recently reported) [5]. It took 45 min (including the CFR assessment). Color-guided blood flow velocity Doppler recording was performed in the LMCA and in the whole LAD. The LAD was subdivided as previously reported, into the proximal part (retropulmonary segment) extending from the left main coronary artery bifurcation to the plane of the pulmonary valve, the mid part still spatially oriented as the strictly retropulmonary part running beyond the pulmonary valve plane in the interventricular groove along the left border of the anterior wall of the right outflow tract (proximal interventricular portion) [13], and distal portions, mostly vertically-oriented, running in the distal part of the interventricular grove (distal segment) [5].

The flow mapping showed no localized blood flow acceleration in the LMCA and in the proximal LAD (Figure 2, top); in the mid LAD a short segment of aliasing was observed, indicating short, mild lumen narrowing that was quantified as <38% of % area reduction by the continuity equation (Figure 2, bottom); this quantification has been widely validated both in severe [14,15] and mild coronary stenosis [5]. No other segments of acceleration were observed. Then, coronary flow reserve (CFR) was assessed in the distal LAD (Figure 3) in order to measure a poststenotic CFR that is specifically affected by a more proximal located stenosis. In fact, in branching arteries like the LAD a CFR more proximally evaluated could be a prestenotic CFR, and in that case CFR can be pseudo-normalized by the branching arteries between the stenosis and the sampling site [7,16]. CFR was measured, as amply validated, by blood flow velocity color-guided PW Doppler recording in the distal LAD at baseline and during a 3-min i.v. infusion of Adenosine (140 mcg/Kg/m) [8,17] and calculated as the ratio of hyperemic peak diastolic flow velocity to resting peak diastolic flow velocity (Figure 3). The CFR resulted 3.2, ruling out a critical coronary stenosis in the LMCA and in the LAD, and also indicating a good microcirculatory function [18].

The patient then underwent ICA (Artis zee, Siemens Healthineers, Erlangen, Germany) that confirmed the E-Doppler TTE results, indicating only lumen irregularities in the LAD with very mild lumen narrowing in the mid tract, estimated to be <20% of diameter narrowing. No other functional assessment such as fractional flow reserve was performed, given the observation of a so mild angiographic stenosis (Figure 4).

So the patient was discharged with a diagnosis of atypical chest pain and mild coronary atherosclerosis. The creatinine at discharge was 1.04 mg/dL. The CTA management had subjected the patient to a considerable burden of radiation exposure, summing the CTA and ICA radiation, for no diagnostic benefit.

## 3. Discussion

In this case, for the first time a comparison between E-Doppler TTE and CTA in assessing coronary stenosis is reported, that shows that coronary blood flow assessed by E-Doppler TTE trumps coronary anatomy assessed by CTA and most importantly, without exposing the patient to ionizing radiation and iodinated contrast medium.

### 3.1. Coronary E-Doppler TTE

Coronary E-Doppler TTE is a totally non-invasive ultrasound-based technology that has recently demonstrated maximum feasibility (≅100%) in recording blood flow velocity (BFV) in the LMCA and the whole LAD thanks to the combination with lowering the heart rate (<60 b/m), power Doppler-based technology and new tomographic planes [5]. The importance of coronary physiology by means of Doppler recording of epicardial and transtenotic coronary blood flow velocity was well known since three decades ago [19,20,21]; but such velocities recording has progressively gained clinical relevance only thanks to the increase feasibility of the transthoracic Doppler method by means of technical advances [7,8,14]. These technical advances are crucially important since coronary flow, and in particular, transtenotic coronary flow velocity are difficult signals to record, being weak intensity and low velocity signals coming from moving coronaries that can be only intermittently insonified, so easily cluttered by the high intensity tissue noise (both cardiac and lung) [7]. The technical advances used in this case and recently validated [5], take over both the previously proposed ones based on contrast enhancement along with second harmonic Doppler technology [14], and the more popular technique based on having a good Doppler technology (i.e., the Doppler module present in the machines Sequoia, Siemens, or in Vivid 7, GE) set basically with a low pulse repetition frequency that, however, works only in very selected “good chest” with relatively low heart rate [22,23,24].

#### 3.1.1. Coronary Flow Mechanics

Now, with the last technical advances [5], LMCA and LAD flow and elevated blood flow velocity at the stenosis site can be easily Doppler recorded and the continuity equation applied to assess the % of lumen reduction, a highly validated approach in both the experimental and the clinical arena [5,14,15,25]. In addition, in accordance with confined jet physics, the accelerated transtenotic coronary flow velocity, being the first adjustment to keep the flow rate constant, causes major pressure dissipation as a consequence of viscous and expansion losses [6]. Therefore, the trans-stenotic pressure gradient that functionally characterizes a stenosis, rises. Thus, BFV acceleration (convective acceleration) at the stenosis site is strongly correlated with the pressure gradient across the stenosis (although it is not the only factor that generates the pressure gradient, contrary to free jet models where the friction is negligible) [6]. In fact, in a confined jet, like in coronary stenosis, the pressure drop is related not only to the jet velocity (convective acceleration ) in a quadratic fashion but also the geometry of the orifice (length and severity of the stenosis) and fluid properties (viscosity), all factors that increase the viscous frictions [6]. Nevertheless, the velocity of the jet is strictly related to the pressure drop since it depends on the minimal cross-sectional area and the minimal cross-sectional area within the stenosis, for any given level of flow, is the single most important determinant of stenosis resistance which appears as a second order term in both viscous and separation losses [26].

For this reason, BF acceleration at the stenosis site is, as previously demonstrated, a stenosis-specific parameter that is strongly correlated with fractional flow reserve, i.e., the assessment of the transtenotic pressure drop during hyperemia, considered the gold standard to assess the physiologic significance of a coronary stenosis [27]. Thus, using E-Doppler TTE as opposed to CTA, we directly weighed up the functional significance of a stenosis with no cumbersome morphologic evaluation of the narrowing effect of the plaque on the lumen [28]. In our patient, we found only a mild acceleration in the mid LAD, reflecting a mild stenosis [5].

#### 3.1.2. Coronary Flow Reserve

Moreover, to complete the functional evaluation of the LMCA/LAD we assessed the coronary flow reserve in the distal LAD, optimally achieved with E-Doppler TTE [7,8,11]. This highly validated method is recommended in the ESC guidelines for chronic CAD management [12] and can integrate the effect not only of the stenosis, with all its geometric impact, but also of multiple stenoses (that may be partially missed) and any diffuse disease affecting that part of the coronary tree proximally located to the Doppler sampling (namely the LMCA and proximal and mid LAD) [16]. However, the CFR can be reduced not only by stenosis but also by microcirculatory dysfunction and hemodynamics [29]. However, thanks to this two-step approach (LMCA/LAD mapping plus CFR in the distal LAD), recently validated [11], it is now possible for the first time with E-Doppler TTE to properly interpret the low CFR, understanding whether it is due to stenosis (in which case a significant acceleration will be detected in the LMCA/LAD) or to microcirculatory dysfunction (when there will be either no local acceleration or localized mild BF velocity acceleration in the LMCA and LAD). When the CFR is >3 (as in our patient) the interpretation is straightforward: there was no flow limiting stenosis and a microcirculatory function was normal. In fact, as previously demonstrated with intracoronary Doppler flow wire, microcirculatory dysfunction can be predicted in the absence of critical stenosis, by a CFR below the validated threshold value of 2.36 [30]. In our case microcirculatory dysfunction was absent, being the CFR = 3.2.

Microcirculation dysfunction, as recently reviewed, can be due to several conditions that can exert oxidative stress on endothelium with the consequent reduced bioavailability of nitric oxide and reduced activity of coronary ions channels ending in reducing both vasodilation and CFR [31,32]. These factors are hypertension, diabetes mellitus, tobacco smoke, oxidized low density lipoproteins, pesticide, heavy metals, electromagnetic field, viruses, alcohol and ionizing radiation to name some [33]. Atherosclerosis with the associated paradoxical vasoconstriction, is in part the extreme consequence of the previously listed factors [34]. Moreover coronary microvascular dysfunction can be worsened by myocardial scar and fibrosis (i.e., in cardiomyopathy, myocardial infarction etc.); in these cases CFR is reduced since the fibrosis-induced rarefaction of capillaries, significantly enhances the resistance during hyperemia (the capillaries are the main resistance during hyperemia and since are resistance in parallel the lower the number the higher the resistance) [34]. As the rarefaction of capillaries is commonly compounded with endothelial dysfunction, the resulting CFR is severely impaired in these cases [35]. Our patient had heavy risk factors for functional microcirculatory dysfunction (diabetes, smoke) but also mild epicardial atherosclerosis; nonetheless he did not have CFR impairment showing the importance of genetic predisposition and the need to rely upon the CFR assessment to evaluate microcirculatory dysfunction and ischemic heart disease.

#### 3.1.3. LAD Stenosis Clinical Impact

Although the CC-Doppler TTE evaluation is limited to the LMCA and LAD, the evaluation offers insight into the global coronary atherosclerosis status since coronary atherosclerosis is a diffuse process and, in line with autopsy and intravascular ultrasound findings [36], the LAD is the first coronary to be involved due to its multi-branching structure that favors atherosclerosis [37]. Moreover, LAD stenosis has a major, independent prognostic impact [38]. This explains why a normal E-Doppler TTE examination of LMCA and LAD virtually excludes significant diseases of the circumflex and of the right coronary artery, as recently preliminarily demonstrated [5,39]. If a mild LAD stenosis is detected, as in this case, critical diseases of the circumflex and of the right artery (present in less than half of the cases) [39] should be checked for, employing a high sensitivity ionizing radiation-free test. Peak exercise echocardiography [40] has been working well in our laboratory for this purpose. However, this examination was not performed in our patient since he was already scheduled for coronary angiography, given the results of CTA. Moreover, our highly feasible method can be easily integrated with coronary endothelial function tests (cold pressure test, nitrates) in order to gain unique prognostic information that could previously only be obtained by invasive means [41]. In this patient, with diffuse mild atherosclerosis a cold pressure test could have added in term of prognosis [42].

#### 3.1.4. E-Doppler TTE Limitations

E-Doppler TTE may miss eccentrically remodeled plaque; but this actually rarely happens. In this respect coronary flow can be integrated by evaluating proximal LAD wall thickness using high-resolution trans-thoracic echocardiography [43]. In additions E-Doppler TTE can have problems for stenosis severity assessment in very tortuous small coronaries and may mildly overestimates the severity of non-critical plaques by the continuity equations partially because the effective area of flow through the stenosis is somewhat less than the cross-sectional area of the narrowing, the so called “vena contracta” effect [44]. A low heart rate (<60 b/m) is mandatory especially in difficult chest associated with low velocity flow. Finally the CFR assessed with this method, that is technically a velocity flow reserve, can underestimate the true volumetric flow reserve but this has minor clinical impact as amply demonstrated [45].

### 3.2. Computed Tomography Angiography

CTA is a non-invasive method that has been spreading like wildfire in recent years thanks to rapid evolving technological advances. Impressive 3D coronary imaging is attained. However, the method has major limitations; first of all, it allows only a luminological evaluation, that is limited even with invasive CA, the gold standard for luminology. In fact, there is no clinically reliable prediction of coronary physiology by coronary anatomy assessment, and coronary physiology trumps coronary anatomy for superior clinical outcomes as largely demonstrated [28]. However, coronary anatomy delineation by CTA is even worse than that of invasive CA owing to the inherently limited spatial and temporal resolution of CTA as compared to invasive CA [4]. In our case, CTA provided a significant over-estimation of the grade of the stenosis of the LAD but also the right coronary artery as opposed to the mild atherosclerotic involvement confirmed by the invasive CA. The atherosclerosis of the circumflex, not quantified due to technical limitations of imaging with CTA, was only mild according to the invasive CA. The cause of the false positives was blooming artifacts due to calcification and perhaps also a mild movement artifact as the heart rate was still >60 b/m in our patient even after B-blocker therapy [46]. Recently, with the introduction of wide detector CT, such as 256- or 320-row, the technique has been further improved thanks to the increased detector rows and improved temporal resolution. However, despite all these technological advances, the specificity and positive predictive value are still limited and not significantly improved as compared to 64-row multidetector computed tomography (MDCT) [46].

#### CTA Risk

Moreover, CTA is based on the use of ionizing radiation and iodinated contrast medium, both of which can potentially seriously damage the human organism. In particular, even radiation exposure <10 msSv, that is the minimal possible radiation exposure with modern CTA equipment, and not guaranteed as there are the huge differences among different laboratories [47], can provoke major damage, as recently demonstrated [48]. CTA radiation exposure causes first of all DNA damage (carcinogenic effect) but such damage takes several years of incubation before reaching the clinical horizon; it has been recently demonstrated that medical radiation like fluoroscopy for coronary assessment (whose radiation exposure is similar to CTA) causes increased mortality for cancer after 15 years from exposure [49]. But CTA also causes immediate lymphocytes death, with a possible involvement of the naïve CD8+ T lymphocytes, a limited population of pluripotent cells devoted to immunity [48]. These are the only cells that can mount an immunological reaction versus novel pathogens; so especially now in the havoc of the COVID-19 pandemic, this kind of cells is monumentally important and their elimination through radiation exposure could be catastrophic [50]. In addition, low dose medical radiation by means of CT and fluoroscopy can cause destabilization of coronary plaque, becoming an essential cofactor in causing acute myocardial infarction. According to John Gofman, there is evidence that more than 50 percent of deaths from cancer and more than 60 percent of deaths from ischemic heart disease may be x-rays-induced by CT and fluoroscopy, being radiations an essential co-factor in causing cancer [51]. As radiation exposure is additive, if CTA requires further examinations involving radiation exposure like in our case, a further burden of radiation exposure by ICA is added. In our case, the patient underwent 11.5 mSv by CTA and an estimated 7 mSv from fluoroscopy, yielding a total burden of ≅20 mSv. Such radiation exposure significantly increases cancer risk [52] especially in consideration that ionizing radiation genotoxic effect adds up to other DNA oxidative damage exerted by other numerous genotoxic agent present in the modern environment like pesticide, electromagnetic field, heavy metals, industrial solvent (toluene, benzene etc.) to name some [33]. In addition, CTA requires the use of iodinated contrast medium, that is highly nephrotoxic. Iodinated contrast media can cause acute kidney injury or be responsible for worsening chronic kidney disease [3]. Such contrast-induced renal toxicity apparently did not happen to our patient as the creatinine remained normal and stable at the discharge.

## 4. Conclusions

The non-invasive assessment of coronary stenosis is of the utmost importance, especially in the light of the ISCHEMIA trial [53], that has reduced the role of PTCA in the management of patients with stable coronary artery disease. E-Doppler TTE, thanks to recent technical advances, can reliably allow an advanced and totally non-invasive assessment of coronary stenosis affecting the LMCA and LAD, based on the functional evaluation of the stenosis without exposing the patient to ionizing radiation. This has major clinical and prognostic implications and promises to be superior to the luminology evaluation of the CTA, that also exposes the patient to ionizing radiation and iodinated contrast medium. Further comparative studies between E-Doppler TTE and CTA are urgently needed to more soundly establish their respective role in the clinical arena.

## Figures and Tables

**Figure 1 diagnostics-11-00245-f001:**
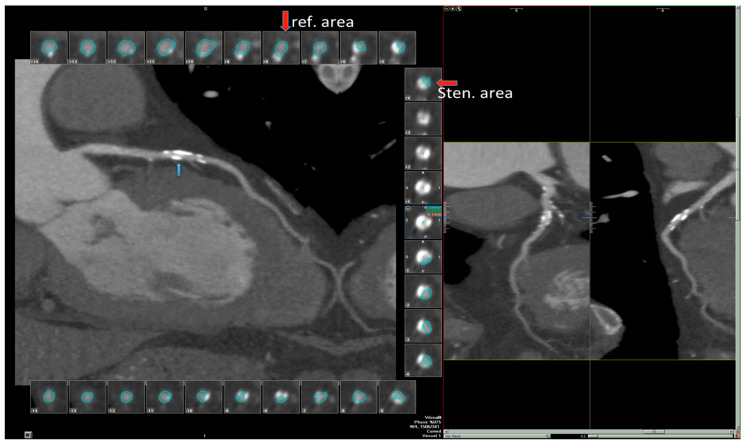
Angiography computed tomography (CTA) of left main coronary artery (LMCA) and left anterior descending coronary artery (LAD). Obstructing plaque is evident in the longitudinal (**blue arrows**) and short axis view (**red arrows**) where the lumen is automatically traced: the most restricted area is indicated as stenotic area (sten area, **red arrow**) and the reference segment as ref area (**red arrow**). The plaque appears hyperreflective (for possible calcification). The stenosis appear critical (>75% area narrowing). Sten = stenosis; ref = reference.

**Figure 2 diagnostics-11-00245-f002:**
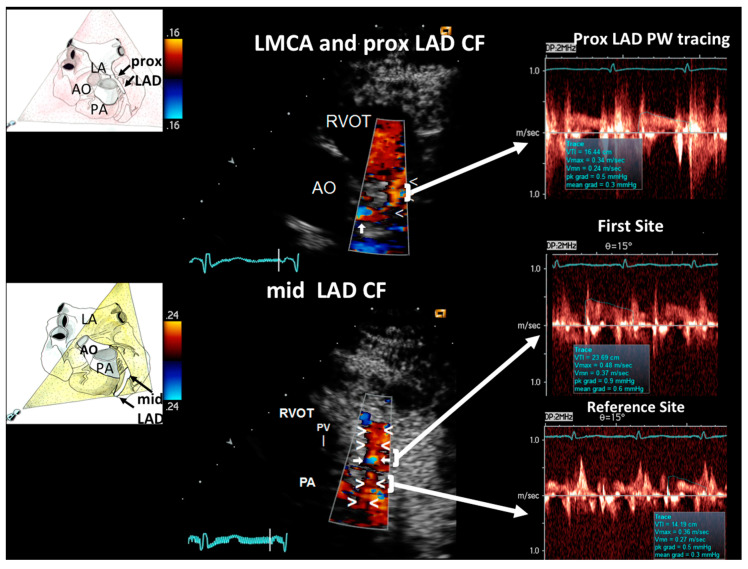
Blood flow Doppler recording by E-Doppler TTE in LMCA and LAD (proximal and mid tract). At the top, color flow in the LMCA and proximal LAD with the corresponding pulsed wave (PW) Doppler tracing in the proximal LAD at the right (arrow indicates the area of sampling: the flow is pretty normal). At the bottom color flow in the mid LAD (center, indicates by arrowheads) that shows a limited area of aliasing (arrows); some flushing artifacts are also present since the heart rate was suboptimally reduced; they do not disturb the proper signal interpretation though. PW Doppler sampling (at the right) at the color aliased signal, confirms mild increase of blood flow velocity with respect to the proximal reference area (indicated by the lower arrow). The application of the continuity equation gives a % stenosis area of 28%. A cartoon to indicate plane orientation is at the left of each color Doppler image. The outline of prevalent diastolic wave of the Doppler tracings is traced in blue. LMCA = left main coronary artery; LAD = left anterior descending coronary artery; PW = pulsed wave; CF = coronary flow; RVOT = right ventricular outflow tract; PV = pulmonary valve; PA = pulmonary artery AO = aorta; LA = left atrium.

**Figure 3 diagnostics-11-00245-f003:**
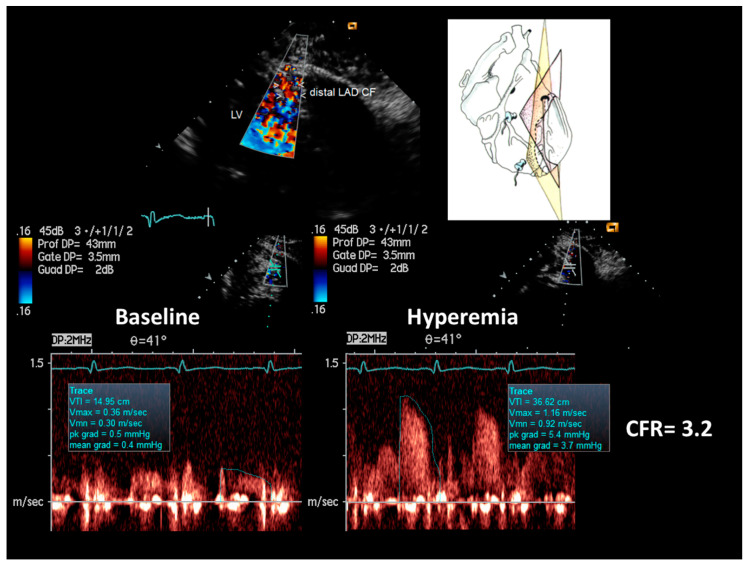
Peak hyperemic diastolic velocity/peak rest diastolic velocity (CFR) in the distal LAD by E-Doppler TTE. At top color flow in the distal LAD (in red); on the right cartoon of the tomographic plane orientation to get such LAD insonification. At the bottom PW spectral tracing of blood flow velocity in the distal LAD at the baseline (**left**) and at maximal Adenosine-induced hyperemia (**right**); notes the prevalent diastolic blood flow velocity, peculiarity of coronary flow. The CFR (peak hyperemic diastolic velocity/peak rest diastolic velocity) is above three, indicating no significant stenosis and normal microcirculatory function. LV = left ventricle; CF = coronary flow.

**Figure 4 diagnostics-11-00245-f004:**
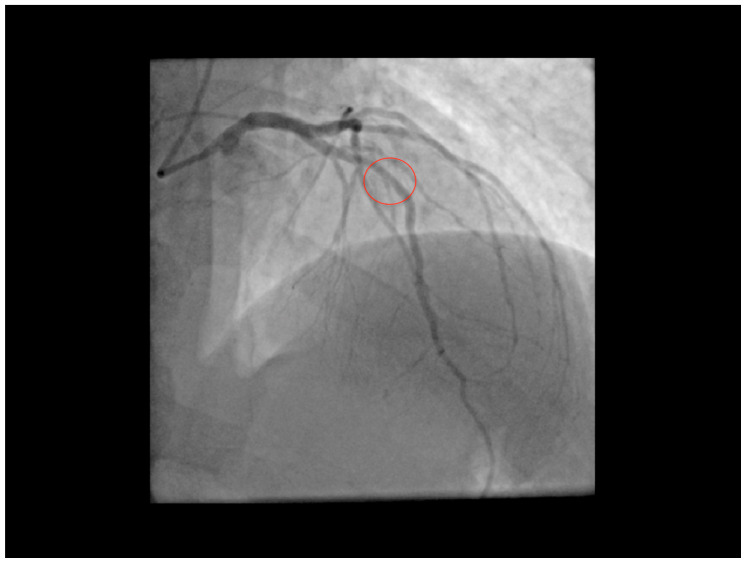
Coronary angiography of the left coronaries (CRA31 view). Coronary angiography appears without evident narrowings, but showing only subtle intimal irregularities; in particular, in the mid LAD (**red circle**) a minimal mild stenosis is evident. CRA31 = antero-posterior view with 31° cranial angulation.

## Data Availability

Data is contained within the article.

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
