# Peer review of "Coronary Flow and Reserve by Enhanced Transthoracic Doppler Trumps Coronary Anatomy by Computed Tomography in Assessing Coronary Artery Stenosis"

_diagnostics, 2021, doi:10.3390/diagnostics11020245_

Round 1

Reviewer 1 Report

This is a case report of a 71-year-old patient who suffered from atypical chest pain. CTA revealed a stenotic lesion of >50% in the proximal LAD, and the patient was scheduled for invasive coronary angiography (ICA). Before ICA he underwent E-Doppler TTE, which noninvasively revealed only mild stenosis of the mid LAD and a CFR of 3.20. In agreement with E-Doppler TTE findings, ICA revealed only a very mild stenosis in the mid LAD and mild atherosclerosis in the other coronaries.

While the topic is of interest, there are some concerns with this report.

This is a typical case of over-estimation of the degree of the stenosis due to calcification by CCT. However, the authors have not described how E-Doppler TTE should be used in detail. Do they believe that E-Doppler TTE can completely exclude the possibility of significant stenotic lesions in patients with positive CCT results? Do they believe that noninvasive E-Doppler TTE should be performed prior to CCT?

As the authors described, noninvasive evaluation is ideal, and TTE is able to assess coronary stenosis in some cases. However, in other cases, it is impossible to visualize lesions clearly during all the cardiac cycle. Moreover, in some cases, we cannot exclude that the angle between coronary blood flow and Doppler beam during velocity measurements could be incorrect, leading to errors in velocity measurements. E-Doppler TTE assessment of coronary stenosis is sometimes useful, but there are so many exceptions. E-Doppler TTE is not always so much reliable even with highly advanced technologies described in the manuscript to exclude the possibility of significant coronary stenosis in patients with chest pain and positive CCT findings.

At least, in this report, E-Doppler TTE results should be validated by other modalities such as adenosine-stressed CMR or flow wire-study.

Author Response

Reviewer#1: This is a typical case of over-estimation of the degree of the stenosis due to calcification by CCT. However, the authors have not described how E-Doppler TTE should be used in detail. Do they believe that E-Doppler TTE can completely exclude the possibility of significant stenotic lesions in patients with positive CCT results?

Response#1: We thank the reviewer for this comment. We think that E-Doppler TTE can very reliably assessed a stenosis and quantify its severity along LAD and LMCA. The stenosis can be measured in terms of length and velocity acceleration (transtenotic velocity acceleration): that latter more crucial point first by color and then by pulsed wave Doppler for a precise quantitative evaluation. This local acceleration can be cought with high accuracy both for critical and subcritical stenosis. In one recent report presented at the ACC 2020 (virtually, for the Covid pandemic) in 73 patients with a coronary flow reserve (CFR)<2.0, a critically accelerated stenotic flow (max diastolic velocity > 93cm/s ) in the LAD/LMCA had a sensitivity of 94% and specificity of 100% versus angiography in detecting a critical stenosis. The 3 patients missed (intermediate stenosis at coronary angiography) had in any case an acceleration detected but that was not that big to give a critical stenosis by the application of the continuity equation; in these 3 cases we think that the CFR was so low no for the underestimation by E-Doppler TTE of the angiographic stenosis but for an associated diffuse coronary atherosclerosis. That was confirmed by IVUS in 2 of the 3 cases. So in some cases you can perhaps underestimate the severity but never totally miss a real critical stenosis. But the real strength of the method is that it can be combined with a coronary flow reserve assessment in the distal LAD that is a highly reliable and validated method for coronary reserve assessment that can indirectly confirm or disconfirm the physiological significance of an epicardial stenosis; in a branching system like LAD only a very distal CFR can very precisely reflect the pressure drop at the level of a stenosis more proximally located. So in the absence of a blunted CFR (>3.0) no critical stenosis can be present. In case of a blunted flow reserve the method can assess critical, mild and as preliminarily demonstrated using intracoronary ultrasound as gold standard, even a diffuse athero (data submitted to ACC 2021). In fact we have noted that a low CFR with a mild local accelerated stenotic flow or a diffuse accelerated flow (>50 cm/s) in the absence of LVH and anemia can be associated to a diffuse atherosclerosis as confirmed by IVUS. Diffuse athero can critically dampen hyperemic flow as amply demonstrated in PET studies. Contrary an impaired CFR with a totally absent acceleration along with a relatively low resting velocity values (<50 cm/s), can indicate a pure microcirculation dysfunction.

Reviewer#2: Do they believe that noninvasive E-Doppler TTE should be performed prior to CCT?

Response#2: Thanks for this question. In this case E-Doppler TTE was performed after CTA for clinical reasons. In a systematic comparative study I think that the most important point is to perform the gold standard method (in this case coronary angiography) after the tested methods namely CTA and E-Doppler TTE as we did in this case .

Reviewer#3: As the authors described, noninvasive evaluation is ideal, and TTE is able to assess coronary stenosis in some cases. However, in other cases, it is impossible to visualize lesions clearly during all the cardiac cycle. Moreover, in some cases, we cannot exclude that the angle between coronary blood flow and Doppler beam during velocity measurements could be incorrect, leading to errors in velocity measurements. E-Doppler TTE assessment of coronary stenosis is sometimes useful, but there are so many exceptions. E-Doppler TTE is not always so much reliable even with highly advanced technologies described in the manuscript to exclude the possibility of significant coronary stenosis in patients with chest pain and positive CCT findings.

Response#3: Thank you very much for this comment. As we said before this method is so incredibly sounder now thanks to the combination of the reduction of heart rate, sensitive Doppler technologies and new tomographic plane that is really difficult to miss a critical stenosis (see previous answer#1 ). It follows some reproducibility results (inter- intraobserver) recently published [1] that can support the reliability of the method:

Table 2. Inter-observers reproducibility results

Variables

First operator

Second operator

STATISTICS

Mean diff

Intraclass correlation coefficient

K

P value

% stenosis CSA

28%±22

28%±26

0.04 %

0.95

(95%CI 0.79- 0.99)

-

<.0001

% increment of velocity

44% ±43

48%±57

-4%±37

0.84

(95%CI 0.37-0.96)

=. 005

LAD color flow global length

92.22 ±12 mm

88.56 ±12 mm

3.4±8 mm

0.88

(95% CI 0.54- 0.97)

-

= .002

Aliasing #

7

7

-

-

1.0

<.0001

Site of aliasing

5 prox; 1 mid; 1 distal

5 prox; 1 mid; 1 distal

-

-

1.0

<.0001

Theta angle correction

17.9°

17.1°

0.8±8°

0.93

(95% CI 0.69- 0.98)

-

= .001

Duration of examination (min)

15±2

14±2

1.7±3.4 min

0.19

-

ns

Mean diff = mean difference; LAD= left anterior descending coronary artery; aliasing # = number of LAD segments with aliased color signal; prox= proximal; K= K measure of agreement; CSA = cross sectional area; CI= confidence interval; min= minutes; ns= not significant

As you can see also the theta angle correction is highly reproducible and is pretty small (<30°) so protecting from major errors. But the major strength of the method is that the accelerated stenotic flow could be corroborated by the super validated CFR; so the combination of the two is really a superior tool to functionally assess stenosis in the LAD and LMCA.

Of course in a comparison with another validated method like CTA we need a gold standard. And in this case we used invasive coronary angiography that was almost normal; so the gold standard confirmed the data of E-Doppler TTE and disconfirmed those of CTA regarding stenosis severity assessment.

However we added in the manuscript also the limitations of this Doppler method (in red page 8).

Reviewer#4: At least, in this report, E-Doppler TTE results should be validated by other modalities such as adenosine-stressed CMR or flow wire-study.

Response#4: Thanks again for your last comment. However we used the most reliable gold standard: coronary angiography that was pretty normal so hampering further more invasive evaluation in this patient (like FFR etc..). If you think that CFR assessment by E-Doppler TTE is not so precise, I have to say that that approach has been already amply validated versus intracoronary Doppler flow wire several years ago [2]. And the two approaches look like really interchangeable (r >0.9) . On the other hands adenosine-stress CMR is a novel technique that has been validated in a few patients versus angiography. So in my view it can’t have the role of gold standard now but I agree that MRI method can be used for comparative studies in the future.

  1. Caiati C, Lepera ME, Pollice P, Iacovelli F, Favale S. A new noninvasive method for assessing mild coronary atherosclerosis: transthoracic convergent color Doppler after heart rate reduction. Validation vs. intracoronary ultrasound. Coron Artery Dis. 2020;31(6):500-11.
  2. Caiati C, Montaldo C, Zedda N, Montisci R, Ruscazio V, Lai G, et al. Validation of a new noninvasive, method (contrast-enhanced transthoracic second harmonic echo Doppler) for the evaluation of coronary flow reserve - Comparison with intracoronary Doppler flow wire. J Am Coll Cardiol. 1999;34(4):1193-200.

Reviewer 2 Report

This is a very interesting case report.

The authors report the case of a 71-year-old patient with many risk factors for coronary atherosclerosis, who, after an episode of atypical chest pain, was diagnosed with critical stenosis of the proximal anterior descending coronary (LAD) by computed coronary angiography (CTA).

The subsequent invasive coronary angiography (ICA) revealed, however, only a very mild stenosis in the mid LAD and mild atherosclerosis in the other coronaries. Enhanced transthoracic echo -Doppler (E-Doppler TTE) of the left main (LMCA) and whole LAD, performed before ICA, found out only mild stenosis of the mid LAD and a normal coronary flow reserve. Thus, coronary stenosis was better predicted by E-Doppler TTE than by CTA.

The authors underscore that, thanks to technical advances in ultrasound imaging, coronary E-Doppler TTE is a totally non-invasive tool for assessment of hemodynamic significance of coronary stenosis, although limited to the LMCA and LAD. The functional assessment instead of an anatomy based  approach makes E-Doppler TTE superior to CTA that also exposes the patient to risks related to the use of ionizing radiation and iodinated contrast medium.

The data are clearly presented and properly discussed. The figures are consistent.

Author Response

Thanks very much for your appreciation.

Reviewer 3 Report

This case report is interesting and well written. It demonstrates the feasibility a superiority of coronary flow and CFR detemination, as well as coronary stenosis definition, using E-Doppler TTE instead of CTA. However i would suggest some minor revision to improve your manuscript:

1) Further english editing is required

2) please define all the abbreviation first time they appear in the text, abstract, figures and tables.

3) Microvascular dysfunction as possible mechanism of ischemic heart disease is only cited in the text and only according with the possible role of echocardiography for its diagnosis. Please further discuss the pathophysiological mechanisms involved in coronary microvascular dysfunction (Int J mol Sci 2020 Oct 30;21(21):8118. doi: 10.3390/ijms21218118.---Int J Mol Sci 2020 Apr 30;21(9):3167.doi: 10.3390/ijms21093167.---- J Am Coll Cardiol 2019 Nov 12;74(19):2350-2360.doi: 10.1016/j.jacc.2019.08.1056).

4) the title is too long and redundnat. Try to find a shorter title, more concise, in relation with the topic of the text.

5) please define the possible limitation of E-Doppler TTE approach in the definition of coronary stenosis.

Author Response

Reviewer#1: Further english editing is required

Response#1: We thank the reviewer for this appropriate comment. We tried to improve the English further.

Reviewer#2: please define all the abbreviation first time they appear in the text, abstract, figures and tables.

Response#2: We thank the reviewer for this appropriate comment. We checked all the abbreviations and found some abbreviations with no definitions (ESC, CMR, ROI, CAD); the definitions were added (in red in the manuscript) page 2; we took out the abbreviations HR , AKI, CKD and used the extensive definitions only : “heart rate” , “acute kidney injury “, “chronic kidney disease “ respectively in order to improve readability being those abbreviations cited only once.

Reviewer#3: Microvascular dysfunction as possible mechanism of ischemic heart disease is only cited in the text and only according with the possible role of echocardiography for its diagnosis. Please further discuss the pathophysiological mechanisms involved in coronary microvascular dysfunction (Int J mol Sci 2020 Oct 30;21(21):8118. doi: 10.3390/ijms21218118.---Int J Mol Sci 2020 Apr 30;21(9):3167.doi: 10.3390/ijms21093167.---- J Am Coll Cardiol 2019 Nov 12;74(19):2350-2360.doi: 10.1016/j.jacc.2019.08.1056).

Response#3: We thank the reviewer for this appropriate comment. We have added comments on mechanism involved in coronary microvascular dysfunction (page 7-8, in red):” Microcirculation dysfunction, as recently reviewed, can be due to several conditions that can exert oxidative stress on endothelium with the consequent reduced bioavailabilty of nitric oxide and reduced activity of coronary ions channels ending in reducing both vasodilation and CFR [31,32]. These factors are hypertension, diabetes mellitus, tabacco smoke, oxidized low density lipoproteins, pesticide, heavy metals, electromagnetic field, viruses, alcohol and ionizing radiation to name some [33]. Atherosclerosis with the associated paradoxical vasoconstriction, is in part the extreme consequence of the previously listed factors [34]. Moreover coronary microvascular dysfunction can be worsened by myocardial scar and fibrosis (i.e. in cardiomyopathy, myocardial infarction etc); in these cases CFR is reduced since the fibrosis-induced rarefaction of capillaries, significantly enhances the resistance during hyperemia (the capillaries are the main resistance during hyperemia and since are resistance in parallel the lower the number the higher the resistance) [34]. As the rarefaction of capillaries is commonly compounded with endothelial dysfunction, the resulting CFR is severely impaired in these cases [35]. Our patient had heavy risk factors for functional microcirculatory dysfunction (diabetes, smoke) but also mild epicardial atherosclerosis; nonetheless he did not have CFR impairment showing the importance of genetic predisposition and the need to rely upon the CFR assessment to evaluate microcirculatory dysfunction and ischemic heart disease.

Reviewer#4: the title is too long and redundnat. Try to find a shorter title, more concise, in relation with the topic of the text.

Response#4: We thank the reviewer for this appropriate comments. We have shortened the title (25% words less): “Coronary Flow and Reserve by Enhanced Transthoracic Doppler Trumps Coronary Anatomy by Computed Tomography in Assessing Coronary Artery Stenosis.”

Reviewer#5: please define the possible limitation of E-Doppler TTE approach in the definition of coronary stenosis.

Response#5: We thank the reviewer for this appropriate comments. We have added these comments (in red, page 8): “E-Doppler TTE limitations. E-Doppler TTE may miss eccentrically remodeled plaque; but this actually rarely happens; in this respect coronary flow can be integrated by evaluating proximal LAD wall thickness using high-resolution trans-thoracic echocardiography [8]. In additions E-Doppler TTE can have problems for stenosis severity assessment in very tortous small coronaries and may mildly overestimates the severity of non-critical plaques by the continuity equations partially because the effective area of flow through the stenosis is somewhat less than the cross-sectional area of the narrowing, the so called “vena contracta” effect [9]. A low heart rate (<60 b/m) is mandatory especially in difficult chest associated with low velocity flow. Finally the CFR assessed with this method, that is technically a velocity flow reserve, can underestimate the true volumetric flow reserve but this has minor clinical impact as amply demonstrated [10]”.

Submission Date

22 December 2020

Date of this review

19 Jan 2021 18:01:29

  1. Severino P, D'Amato A, Pucci M, Infusino F, Birtolo LI, Mariani MV, et al. Ischemic Heart Disease and Heart Failure: Role of Coronary Ion Channels. Int J Mol Sci. 2020;21(9).
  2. Severino P, D'Amato A, Pucci M, Infusino F, Adamo F, Birtolo LI, et al. Ischemic Heart Disease Pathophysiology Paradigms Overview: From Plaque Activation to Microvascular Dysfunction. Int J Mol Sci. 2020;21(21).
  3. Caiati C, Pollice P, Favale S, Lepera ME. The Herbicide Glyphosate and Its Apparently Controversial Effect on Human Health: An Updated Clinical Perspective. Endocr Metab Immune Disord Drug Targets. 2019.
  4. Caiati C. Contrast-Enhanced Ultrasound Reveals That Lipoprotein Apheresis Improves Myocardial But Not Skeletal Muscle Perfusion. JACC Cardiovasc Imaging. 2019;12(8 Pt 1):1441-3.
  5. Tsagalou EP, Anastasiou-Nana M, Agapitos E, Gika A, Drakos SG, Terrovitis JV, et al. Depressed coronary flow reserve is associated with decreased myocardial capillary density in patients with heart failure due to idiopathic dilated cardiomyopathy. Journal of the American College of Cardiology. 2008;52(17):1391-8.
  6. Gradus-Pizlo I, Sawada SG, Wright D, Segar DS, Feigenbaum H. Detection of subclinical coronary atherosclerosis using two-dimensional, high-resolution transthoracic echocardiography. Journal of the American College of Cardiology. 2001;37(5):1422-9.
  7. Caiati C, Lepera M, Pollice P, Favale S. Feasibility of a New Non Invasive Method for the Evaluation of Coronary Blood Flow in Coronaries: Transthoracic Convergent Color Doppler Mode along with Pharmacologically Induced Heart Rate Lowering. J Am Coll Cardiol. 2020;75(11 Supplement 1):1786.
  8. Caiati C, Montaldo C, Zedda N, Bina A, Iliceto S. Assessment of coronary flow reserve by contrast-enhanced second harmonic echo Doppler - Response. Circulation2000. p. E100-E.

Round 2

Reviewer 1 Report

The authors appropriately addressed all issues.

I have no further comment.